# The Impact of White Mulberry, Green Barley, Chia Seeds, and Spirulina on Physicochemical Characteristics, Texture, and Sensory Quality of Processed Cheeses

**DOI:** 10.3390/foods12152862

**Published:** 2023-07-27

**Authors:** Monika Garbowska, Anna Berthold-Pluta, Lidia Stasiak-Różańska, Stanisław Kalisz, Antoni Pluta

**Affiliations:** 1Division of Milk Technology, Department of Food Technology and Assessment, Institute of Food Sciences, Warsaw University of Life Sciences—SGGW, Nowoursynowska 159c Street, 02-776 Warsaw, Poland; monika_garbowska@sggw.edu.pl (M.G.); lidia_stasiak_rozanska@sggw.edu.pl (L.S.-R.); antoni_pluta@sggw.edu.pl (A.P.); 2Division of Fruit, Vegetable and Cereal Technology, Department of Food Technology and Assessment, Institute of Food Sciences, Warsaw University of Life Sciences—SGGW, Nowoursynowska 159c Street, 02-776 Warsaw, Poland; stanislaw_kalisz@sggw.edu.pl

**Keywords:** processed cheese, TPA profile, plant-based additives, CIE L*a*b* system

## Abstract

Processed cheeses (PC) are products resulting from the mixing and melting of rennet cheese, emulsifying salts, water, and possibly various additional ingredients. They are considered good vehicles for new ingredients, including plant-based ones. In addition to the health-promoting effects of plant-based ingredients, some of them can also affect positively the quality characteristics of PC (e.g., texture, taste, and consistency) and their addition may reduce the amount of emulsifying salts used. The aim of the study was to determine the possibility of the addition of 0.5, 1.0, 2.0, and 3.0% white mulberry (M), chia (Ch), green barley (GB), or spirulina (S) to PC and the effects on selected characteristics of these products (chemical composition, pH, water activity, color parameters, texture, and sensory properties). In all PC variants, a significance decrease in the dry matter content was observed with an increase in the additive level. The use of plant-based additives allowed us to reduce the addition of emulsifying salts by 50% compared to their typical amounts and the share of rennet cheese in the PC recipe by approximately 18%, which had a beneficial effect on the nutritional value of these products. The use of 3% GB, Ch, or M as additives to PC enabled a reduction in its sodium content by 27, 27, and 42%, respectively, compared to the control cheese. Among the tested additives, GB caused the greatest increase in the hardness of PC (even at the amount of 0.5%), indicating that is beneficial and can be used in the production of sliced PC. All the additives either significantly reduced the adhesiveness of PC or had no effect on this parameter. In terms of sensory characteristics, the highest acceptable addition of GB was 0.5%, and that for S and Ch was 1%, while the addition of M, even at 3%, was assessed very positively. The results of this research may be helpful in the development of new recipes for processed cheeses obtained in industrial conditions.

## 1. Introduction

Processed cheeses are oil-in-water emulsions, consisting of cheese/cheeses, emulsifying salts, water, and other optional ingredients [1,2]. They are produced by grinding cheese at varying stages of ripening, adding emulsifying salts and other dairy and non-dairy ingredients, and heating the mixture under continuous stirring to obtain a homogeneous product with an extended shelf life. Ingredients are mixed and heated for a minimum of 30 s at a minimum temperature of 65.5 °C to produce a homogeneous mixture, which is then packed and cooled [3]. Processed cheeses contain an average of 25% fat, 9% protein, and approximately 3% carbohydrates. Their most unfavorable feature is the high content of phosphorus—approximately 600 mg/100 g. The nutritional value of processed cheese varies significantly depending on the type; the average energy value of one slice (21 g) of processed cheese is 69 kcal, of which 68% comes from fat, 22% from protein, and 9% from carbohydrates [4].

The global PC market is expected to grow from USD 13.5 billion in 2017 to USD 23.2 billion by 2032. The market is dominated by North America (primarily the United States), with approximately a 35% share, and European countries, with a 29% market share (United Kingdom, Germany, and Spain) [5].

Consumers’ eating habits are extremely difficult to change, despite the well-known positive health effects of good eating habits. Therefore, efforts to enrich conventional food products with new ingredients with known health benefits seem highly justified. Such products could elicit certain nutritional benefits or improve the well-being of consumers in ways that do not require significant modifications in their behavior (diet, eating habits, etc.). Improved products could also provide nutrients whose sufficient levels are difficult to provide in an everyday diet. To meet the needs of consumers seeking food products whose consumption offers benefits beyond basic nutrition, it is worthwhile to identify opportunities to produce processed cheeses with additives that enrich them with active ingredients. Processed cheeses with functional ingredients, such as prebiotics, probiotics, antioxidants, or polyunsaturated fatty acids, have already appeared in retail [6,7,8,9].

In recent years, there has been a trend toward the use of plants as sources of various biologically active substances, exerting antioxidant and anti-inflammatory (for example, phenols, anthocyanins, non-anthocyanin flavonoids), anti-carcinogenic (for example, alkaloids, anthocyanins), and anti-diabetic effects [10,11,12]. However, the comprehensive use of a variety of nutrient-rich plants, especially those with fewer culinary and medicinal applications, is still rare, especially in the context of using them as additives in traditional foods.

Food products made from young barley leaves and shoots, otherwise known as green barley or barley grass (*Hordeum vulgare* L.), have recently become very popular. They are available in various forms: in liquid (as an ingredient in juices) or in dry powdered form. The leaves of barley are beneficial to human health when harvested at a young age (usually within 2 weeks after seeding). Green barley is a source of phenolic acids, many vitamins (C, E, and B), chlorophylls (6.62–19.62 mg/g), folic acid, and macro- and micro-elements (calcium, iron, magnesium, potassium, zinc, and copper) [13,14]. Young barley leaves are a rich source of potent antioxidants (saponarins and lutonarins), owing to which they exhibit hypolipidemic, antidiabetic, anti-ulcer, and antidepressant effects [15].

Spirulina (also referred to as Arthrospira) is a blue-green algae (the most important species of which include *Spirulina maxima* and *Spirulina platensis*) belonging to the Oscillatoraceae family. These edible cyanobacteria are classified as a “superfoods” based on several activities, including their antioxidant, antidiabetic, cholesterol-controlling, and insulin-resistance effects [16]. In addition to protein (60–70%), spirulina also contains carbohydrates (15–18%), vitamins (β-carotene, vitamin B1, B2, B12, and E—0.2 and 4 mg/100 g, respectively), minerals (including iron—50–100 mg/100 g, magnesium—200–300 mg/100 g, calcium—400–600 mg/100 g), essential fatty acids (e.g., γ-linolenic acid—GLA—1–2%), and pigments (chlorophylls—1–2%, phycocyanin 8–12%, and carotenoids—350–450 mg/100 g) [17,18]. GLA is a very rare ingredient in everyday diets and is used in the prevention of various diseases and disorders, including atopic eczema, premenstrual syndrome, diabetes, cardiovascular disease, and inflammation [16].

Mulberry, which belongs to the *Morus* genus of the Moraceae family, is an aggregated berry whose fruits have an oval shape and unique taste. There are 24 species in the *Morus* genus and hundreds of varieties. Since ancient times, different varieties of mulberry have been used in Asian countries, both as food (fruit) and in medicine (fruit, bark, leaves) [19]. Mulberry (*Morus alba* L.) fruits are most popular in East Asian countries such as Korea, China, Japan and Thailand, but are also consumed worldwide [20]. Due to the presence of polyphenols (such as phenolic acids (up to ~30 mg/kg fw), flavanol derivatives (from 55.62 to 432.38 mg/kg fw), and anthocyanins (up to ~3000 mg/kg fw)), polysaccharides, alkaloids, flavonoids, essential amino acids, and vitamins, white mulberry fruits are well known for their health-promoting properties, including anticoagulant, antioxidant, anti-diabetic, anti-obesity, anti-inflammatory, anti-carcinogenic, prokinetic, and neuroprotective effects [12,19,20,21,22,23].

*Salvia hispanica* L. is an herbaceous plant of the Lamiaceae family and the genus *Salvia*, which includes approximately 900 plant species found in almost all parts of the world: North and South America, South Africa, Southeast Asia, and Europe [24]. The lipids of chia seeds have high content of polyunsaturated fatty acids (PUFAs) of the omega-3 (α-linolenic acid, 54–67%) and omega-6 (linoleic acid, 12–21%) families, while being low in saturated fatty acids. Chia seeds are also a rich source of protein (15–24%), dietary fiber (both oligosaccharides and polysaccharides, such as cellulose, hemicelluloses, pectic substances, and gums, 30–34%), carbohydrates (26–41%), minerals (especially calcium and magnesium), and vitamins. They also contain bioactive compounds, such as phenolic acids (gallic acid, caffeic acid, ferulic acid, *p*-coumaric acid), depsides (chlorogenic acid, rosmarinic acid), flavonoids, isoflavones, catechin derivatives, tannins, phytates, carotenoids, and sterols [24,25,26,27]. Studies based on in vitro assays and animal and human models have proven that chia seeds exhibit neuroprotective, hepatoprotective, cardioprotective, hypotensive, anti-inflammatory, antioxidant, anti-diabetic, and anti-atherosclerotic properties [24,27]. An important technological feature of chia seeds is their ability to absorb large amounts of water, leading to the formation of a transparent gel called chia mucilage, composed mainly of fiber [26]. Due to these properties, chia seeds are exploited in the food industry as thickeners, emulsifiers, stabilizers, or anti-freezing agents. Because of their high nutritional value, their nutritional and health benefits in the prevention and treatment of lifestyle-related diseases, and also due to technological reasons, they are eagerly incorporated into the recipes of certain food products [25,26]. In 2009, the European Food Safety Authority issued a positive opinion on the safe use of chia seeds in the food industry [28].

In addition to plant-derived ingredients’ health-promoting activity, some of them (for example, chia seeds) can also affect the quality characteristics of processed cheeses, e.g., texture. At the same time, their addition may enable a reduction in the amounts of emulsifying salts used. The aim of the present study was to determine the possibility of using functional ingredients, such as white mulberry, chia seeds, green barley, and spirulina, to produce processed cheeses, and to examine the effects of these additives on their physicochemical composition, pH, water activity, texture, color parameters, and sensory properties.

## 2. Materials and Methods

### 2.1. Experimental Material

The study material consisted of processed cheeses produced under laboratory conditions with the addition of green barley (*Hordeum vulgare* L.), spirulina (*Spirulina platensis*), chia seeds (*Salvia hispanica* L.), and white mulberry fruit (*Morus alba* L.), and cheeses without the plant additives as control samples.

The processed cheeses were produced from Dutch-type rennet cheese with a fat content of 45% in dry matter and butter (Dairy Cooperative Mazowsze in Chorzele, Poland), emulsifying salts (Self^®^ M 9.0 MV6233, Budenheim, Germany), and the following additives:-green barley powder (Intenson Europe Sp. z o.o., Całowanie, Poland; composition per 100 g: protein—29 g, fat—5 g, sugars—3.4 g, dietary fiber—39 g);-spirulina powder (GREEN Essence Sp. z o.o., Konstancin-Jeziorna, Poland; composition per 100 g: protein—60 g, fat—5.9 g, sugars—11 g, dietary fiber—5.1 g);-chia seeds (Sante Sp. z o.o., Warsaw, Poland; composition per 100 g: protein—23 g, fat—34 g, sugars—7 g, dietary fiber—34 g);-dried white mulberry fruits (Purella Superfood, Warsaw, Poland; composition per 100 g: protein—11 g, fat—2.5 g, sugars—59 g, dietary fiber—12 g).

### 2.2. Production of Processed Cheeses under Laboratory Conditions

First, appropriate amounts of individual ingredients included in the melting mixture were prepared. The formula of the melting mixture used to produce cheeses with green barley and spirulina is shown in Table 1. Based on the results of preliminary experiments, which demonstrated a significant change in the consistency of processed cheeses enriched with chia seeds and mulberry fruit, the content of rennet ripening cheese was decreased and the water volume was increased in their formulas (Table 2).

Cheese, previously chopped into smaller pieces, was ground 3 times using a Zelmer ZMM4045W food processor (Eurogama Sp. z o.o., Warsaw, Poland). Dried white mulberry fruits were ground using a Krups GX204 kitchen grinder (Eurogama Sp. z o.o., Warsaw, Poland). The weighed portion of the emulsifying salts was mixed in an appropriate volume of water. The prepared and weighed raw materials were transferred to a pilot-scale cheese cooker (STEPHAN UMC 5, ChEF, Saint Cannat, France) to enable the melting process. The cooker was heated using water from a water bath having a temperature of 95 °C, which circulated in a closed circuit in the cooker jacket. The process conditions were the same for each cheese variant, i.e., the melting process was carried out for 12 min at a speed of 2700 rpm to the final temperature of the cheese bulk of 85 °C. Immediately after the cheese bulk had been obtained, the contents of the cooker were poured into packages in the form of plastic pans and 100-mL cylindrical containers. The samples thus prepared were initially cooled at room temperature, placed in a laboratory refrigerator at 6 °C, and stored until analyzed.

### 2.3. Physicochemical Analyses of Processed Cheeses

The dry matter (*DM*), fat, protein, and sodium content was determined in the processed cheese samples according to ISO standard methods: ISO Standard 5534 [29], ISO Standard 3433 [30], ISO Standard 8968-1 [31], and ISO Standard 8070 [32], respectively. Each of the processed cheese variants was analyzed in triplicate, and the results are reported as mean values of these three analyses.

Based on the results of moisture and fat content, the fat content in dry matter (*FDM*) was calculated using Equation (1):(1)$$FDM=\frac{F}{DM}\times 100\;[\%]$$
where *FDM*—fat content of cheese dry matter [%], *F*—fat content of cheese [%], *DM*—dry matter content of cheese [%].

### 2.4. Measurement of Water Activity of the Analyzed Processed Cheeses

The water activity of the processed cheeses was measured using a properly calibrated ROTRONIC HP23-AW-A water activity meter (B&L International Sp. z o.o., Warsaw, Poland) according to the manufacturer’s instructions and Simatos et al. [33]. Each of the processed cheese variants was analyzed in triplicate.

### 2.5. pH Measurements

An OXYGEN METER CP-505 pH meter (ELMETRON Sp. J., Zabrze, Poland) was used to measure the pH of the processed cheeses. The device was calibrated according to the manufacturer’s instructions using special buffer solutions. Samples were prepared by double homogenization in a SEWARD BA 7020 stomacher (VWR International Sp. z o.o., Gdansk, Poland) for 120 s. Each sample contained 10 g of the resulting processed cheese and 20 mL of distilled water. The emulsion thus prepared was transferred to a measuring vessel, its contents were stirred, and then the pH electrode of the pH meter was immersed, and the pH value was read from the meter’s display monitor [34].

### 2.6. Measurement of Color Components L*, a*, and b*

The color measurement was performed using the method described by Grobelna, Kalisz, and Kieliszek [35] on a KONICA MINOLTA CM-3600d spectrophotometer (Konica Minolta, Osaka, Japan) using a 10° observer type, D65 illuminant, CIELAB system in reflected light using a 25.4 mm aperture. Values of three color components, *L**, *a**, *b**, were read during measurements and the total color difference ΔE was calculated compared to the control sample, according to Equation (2) below [36]:(2)$$\Delta E=\sqrt{{\left(\Delta L^{*}\right)}^{2\;}+{\left(\Delta a^{*}\right)}^{2\;}+{\left(\Delta b^{*}\right)}^{2\;}}$$
where ∆*E*—total color difference (−), ∆*L**—difference in lightness between the sample of cheese with an additive and the control sample, ∆*a**—difference in the value of chromatic coordinate *a** between the sample of cheese with an additive and the control sample, ∆*b**—difference in the value of chromatic coordinate *b** between the sample of cheese with an additive and the control sample.

### 2.7. Texture Analysis of Processed Cheeses

The texture analysis of all processed cheeses was carried using a BROOKFIELD CT3 10K texture analyzer (Labo Plus Sp. z o.o. Warsaw, Poland) based on the Texture Profile Analysis (TPA) test, with a TA4/1000 pin included in the equipment of the apparatus. The speed of the measuring head was set at 1.00 mm/s, assuming 20% deformation. Prior to the analysis, the instrument was calibrated according to the manufacturer’s instructions. Each of the produced cheeses was tested three times—each time using a new cheese sample. The samples of the processed cheeses were placed in cylindrical plastic containers, the diameter of which corresponded to the diameter of the mandrel used. The following texture parameters were determined in the TPA test: hardness [N], adhesiveness [mJ], springiness [mm], cohesiveness [−], gumminess [N], and chewiness [mJ] [34].

### 2.8. Sensory Analysis

The organoleptic assessment was performed according to ISO Standard 22935-3 [37] by a team of 15 experienced panelists. The samples of the processed cheeses were coded and brought to room temperature before the assessment, which included the following quality attributes: appearance, aroma, consistency, and taste. The assessment was carried out using an 8-point scale, where a score of 5.0 indicated a very good, flawless product; 4.5 indicated very good, but with an unspecified flaw; 4.0 indicated good, with a perceptible flaw; 3.5 indicated fairly good, but with a pronounced flaw; 3.0 indicated sufficient, with a serious flaw; 2.5 indicated a very serious flaw; 2.0 indicated a sensorially unacceptable product, and 1.0 indicated an inedible product. The results were presented as mean scores for each sensory quality attribute, whereas an overall sensory acceptability score was calculated as the mean of the attributes, using the same weighting factors for each attribute (0.25).

### 2.9. Statistical Analysis

Statistical analysis was performed using the Statistica 13.1 software (StatSoft Polska Sp. z o.o., Kraków, Poland). One-way analysis of variance (ANOVA) was applied at a significance level of α = 0.05. Tukey’s test was deployed to determine the significance of differences between mean values and to identify homogeneous groups. Analyses were also conducted to establish the effect of the vegetable additive level on selected quality attributes of the processed cheeses produced under laboratory conditions.

## 3. Results and Discussion

### 3.1. Chemical and Physicochemical Characterization of Processed Cheeses

Table 3 shows the results of the determination of the chemical composition and physicochemical properties of the processed cheeses. The *DM* content of the samples ranged from 37.86 to 46.00%. It was found that even the smallest tested level of spirulina, mulberry fruit, or chia seeds (0.5%) caused a significant increase in *DM* content compared to the control sample. In all variants of processed cheese, a significant decrease in *DM* content was observed with the increasing amount of additive. The change in the *DM* content can be explained by the increase in the amount of fiber and polyphenols in PC with increasing additives. Fiber and polyphenols (components present in all the analyzed supplements) can interact with proteins, causing the formation of polyphenol–protein complexes. Up to a certain concentration of polyphenols, this mechanism can reduce moisture release, but excess polyphenols increase it [38].

A similar relationship as in the case of the *DM* content was found in PC with green barley, spirulina, and mulberry in relation to the fat content, which resulted from the reduction in fat introduced to the PC together with the basic raw material, i.e., rennet cheese. Only cheeses with chia had higher fat content (0.5% and 1%) or similar content (2% and 3%) compared to the control PC. Chia seeds are characterized by high fat content (34%, see Section 2.1), which resulted in similar fat content in PC with 3% chia addition (22.49%) to the control PC (22.53%).

The protein content of all cheese variants with green barley was significantly higher (from 16.59% to 17.24% depending on the additive level) compared to that of the control cheese (15.05%). A decrease in protein content was noted in the cheeses with other additives along with their increasing levels, especially in the case of cheeses with 3% chia seeds (13.44%) and 3% mulberry fruit (12.62%). This was due to a significant reduction in the proportion of rennet ripening cheese in the formulas of these cheeses, i.e., from 56% *wt/wt* (in the control processed cheese) to 46% *wt/wt* (in the cheese with 3% of the additive). The protein content of the additives used varied, ranging from 11% in mulberry fruit, through 23–29% in chia seeds and green barley, to 60% in spirulina. This indicates that different amounts of protein were introduced into the PC with these additives (maximally approximately 0.18 g/100 g of cheese in the case of 3% spirulina addition), which was, however, much smaller than the “loss” of protein content resulting from the reduction in the amount of rennet cheese in the PC formula. Our data exhibit the same trend as that found in the studies of Alqahtani et al. [38], El-Loly et al. [39], and Aly et al. [40], which used date fruit seeds, date syrup, and bulgur, respectively.

The sodium content of PC decreased statistically significantly along with the amount of the plant additive. Such a decrease in sodium content in processed cheeses, regardless of the additive type, was achieved by a 50% reduction in the proportion of emulsifying salts, which are the main contributors to the high sodium content of processed cheeses. The lowest sodium content (0.61%) was determined in the cheeses with 3% addition of mulberry fruit and 3% addition of green barley or chia seeds (0.77%). For nutritional reasons (excessive sodium and phosphorus content and an unbeneficial calcium/phosphorus ratio in processed cheeses), research has been conducted for years on the possibility of reducing the amounts of emulsifying salts during processed cheese production through, among other methods, the use of hydrocolloids [41,42]. The present study results indicate that the use of certain additives of plant origin (e.g., 3% mulberry) enables a reduction in the sodium content of processed cheeses by approximately 40% compared to control cheeses without additives, by reducing the proportions of emulsifying salts and rennet cheese.

The enrichment of processed cheeses with additives resulted in some changes in their proximate chemical compositions but also in the introduction of components that are normally absent (oligosaccharides, typical components of plant origin, e.g., chlorophylls) or found in limited quantities (e.g., micronutrients) in these products. A study conducted by Tohamy et al. [43] demonstrated that spirulina’s addition to a spreadable cheese analog at a level of 4% increased its selenium content by 7 times, zinc content by 33 times, and iron content by 4 times compared to the control cheese. Algae addition also introduced new components to the spreadable cheese, such as carotenes, chlorophyll a, phycocyanin, and fiber. If one assumes the addition to be 3% and the fiber content of, e.g., chia seeds to be 34%, the dietary fiber content in the processed cheese will be approximately 0.1%.

The pH of the processed cheeses varied depending on the additive and ranged from 6.01 to 5.57. In cheeses with 2 and 3% green barley addition, there was a reduction in pH of approximately 0.1 units compared to the control cheese sample. The results in Table 3 indicate that the pH in the PC with spirulina increased with the amount of added algae. This may have been due to the pH of the *Spirulina platensis* themselves (pH in the range of 8.5–11). A similar pH level in cheeses with the addition of 2–4% spirulina has been reported in the literature [43]. The greatest decrease in pH was noted in the cheeses enriched with mulberry fruit, with 0.5% addition causing a significant decrease in the pH of the processed cheese to 5.81, and 3% addition to a pH 5.60, compared to the control cheese sample (pH 5.96). Previous studies [39,44] have also indicated that the addition of fruit (e.g., in the form of syrup) or walnut paste may cause a slight decrease in pH in PC.

Water activity in the control cheese reached 0.916 on average. It was found that even the lowest additive level used caused the a_w_ to increase significantly compared to the control sample, to values ranging from 0.933 (for 0.5% chia seeds) to 0.944 (for 0.5% spirulina). In the case of the processed cheeses enriched with green barley and mulberry fruit, the a_w_ did not change with the increasing additive level, but was significantly higher compared to the a_w_ of the control processed cheese. The highest a_w_ was determined for the cheeses with 2.0 and 3.0% spirulina addition and the cheese with 3.0% chia seed addition.

### 3.2. Color Parameters of Processed Cheeses

All the additives used affected the color of the PC (Table 4). Their increasing levels resulted in the darkening of the cheeses, as indicated by a decreased value of the *L** color parameter. The extent of changes in color depended on the additive type and level, with the strongest influence observed in the case of mulberry addition and the weakest one in the case of spirulina addition. The values of ∆*E* were determined to establish to what extent the applied additives affected the color and whether this effect was noticeable to the consumer. According to the ΔE value ranges given by Cserhalmi et al. [45], the color difference between treated and untreated samples can be estimated as imperceptible (0–0.5), hardly noticeable (0.5–1.5), noticeable (1.5–3.0), and clearly visible (3.0–6.0). The 0.5% addition of mulberry fruit caused a noticeable change in the color of PC, which could be noticed by a person competent and experienced in the field. Increasing the mulberry level to 1 and 2% caused a change in color that could be noticed by an inexperienced person. However, significant color differences were noted compared to the control sample upon 3% mulberry addition to the cheese formula. In the case of chia seeds, even their 0.5% addition caused a clearly visible change in the color of PC. Increasing the level to over 0.5% caused significant and visible differences in color compared to the control sample, and so did the addition of green barley extract and spirulina, regardless of the additive level.

Fortification led to significant changes in the color of all processed cheeses. It could be seen that the lightness (*L**) values of the cheeses were significantly decreased, regardless of the type of additive. Moreover, Alqahtani et al. [38] observed a darker color (decreasing *L**) of block-type PC with increasing ratios of date fruit seed and storage duration. The PC with the addition of green barley and algae showed negative values of parameter *a**, which decreased with the amounts of these additives (these changes are clearly visible in Figure 1). The green color that characterized the PC even with 0.5% of green barley or spirulina was obviously due to the presence of chlorophylls in these additives. A similar relationship, but deepening of the redness (increasing positive values of parameter *a**), was found in cheeses with the addition of chia or mulberry, which confirmed the results of other researchers [38]. The yellowness (*b**) values of cheese with green barley were significantly increased with the amount of additive. In turn, in the PC with chia, the opposite relationship was observed.

### 3.3. Texture Profiles of Processed Cheese Samples

The textural properties of processed cheese are affected both by its chemical composition (fat content, moisture content, pH) and raw material properties (type of cheese, pH and degree of proteolysis, amount and type of emulsifying salts), but also by the processing conditions (melting and cooling temperature, mixing speed, time of melting, cooling rate, temperature, and storage time) [38]. Figure 2 shows the effect of 0.5, 1, 2, or 3% addition of green barley, spirulina, chia seeds, or mulberry fruit on the textural properties of the processed cheeses, such as hardness, cohesiveness, adhesiveness, gumminess, springiness, and chewiness.

The hardness of processed cheeses with green barley addition ranged from 56.91 N to 65.47 N; however, additive levels of 1, 2, and 3% were observed to have no effect on the values of this texture parameter. Regardless of the additive level, the hardness of these cheeses was ca. 3–4 times higher compared to the control cheeses (15.11 N) and substantially higher compared to cheeses with the other additives tested. The high hardness of the PC with green barley could have been due to the strong water-binding properties of the proteins and polysaccharides contained in this ingredient [38].

In the case of the processed cheeses enriched with spirulina, an increase in the additive level caused an increase in their hardness, with the highest value of hardness, i.e., 32.17 N, noted for the cheese variant with 3% addition. The opposite dependency was noted for the cheeses with chia seeds and mulberry. Their 0.5% addition caused the hardness of the processed cheeses to increase significantly to 20.09 N and 36.86 N, respectively, which was higher compared to the control cheese (15.11 N). In order to reduce the hardness of cheeses with plant additives added at levels of 1–3%, their formulas were modified by decreasing the content of ripening rennet cheese and increasing the water volume compared to the formulas of processed cheeses with green barley and spirulina. It is common knowledge that a decrease in cheese hardness observed along with a water content increase is due to the enhanced hydration of the protein matrix, which weakens the protein–protein interactions and thus favorably plasticizes the matrix [46,47].

Processed cheeses with added green barley were less cohesive (0.22–0.29) than the control cheese (0.59). Increasing the addition of green barley resulted in an increase in the cohesiveness of the cheeses; however, the value of this characteristic was still significantly lower than in the control sample. On the contrary, increasing the addition of chia seeds decreased the cohesiveness of the cheeses. In the case of mulberry, this addition did not affect the cohesiveness of the cheeses (0.54–0.62) compared to the control samples. Regardless of the amount of spirulina added, the cheeses were less cohesive than the control ones. Greater cohesiveness means that the structure of the cheese will not fall apart easily, which is related to the strength of the internal bonds in the cheese structure. Harder PC is usually less cohesive compared with softer PC [46].

Processed cheeses made with spirulina showed a decrease in adhesiveness with an increasing additive level (cheeses with 0.5 and 3% added spirulina—25.73 and 2.67 mJ, respectively) compared to the control sample (43.67 mJ). The addition of 0.5 and 1% of chia seeds also reduced the adhesion of the processed cheeses, while higher additive levels had no effect on these values. On the other hand, the adhesion of processed cheeses produced with the use of mulberry, regardless of the amount added, was similar to that of the control sample and ranged from 44.64 to 50.47 mJ. The high adhesiveness of processed cheeses to various surfaces (e.g., packaging material) is a drawback and may be one of the parameters reducing their consumption, as consumers do not favor products that are difficult to separate from the packaging [46]. To summarize, all the tested additives either significantly reduced the adhesiveness of the processed cheeses or had no effect on the values of this parameter (mulberry addition), which seems to be beneficial from the consumer’s point of view.

The addition of green barley and spirulina to the formulas of processed cheeses caused a significant increase in their gumminess, which increased along with the additive level, compared to the control sample. In the case of chia seeds used as a plant additive, the highest gumminess was found in the processed cheeses with 1% addition. With the additive level increasing, the values of this texture parameter decreased to 13.79 N (2% of chia seeds) and 11.40 N (3% of chia seeds), assuming a value that did not differ from that determined for the control cheese (10.36 N). This means that the gumminess of processed cheeses with the addition of chia seeds significantly decreased with the decrease in the rennet cheese content in the cheese formula, and at the same time with an increase in the water content. Similar correlations were found for mulberry used as a plant additive, with the processed cheese with 3% mulberry being significantly less gummy than the control cheese.

The springiness and chewiness of the processed cheeses with green barley addition increased along with the increasing additive level. Meanwhile, the opposite correlation was observed in the case of spirulina addition, i.e., the higher the algae level in the formula, the lower the springiness and chewiness of the cheese. When chia seeds were used as a plant additive, the highest springiness was noted for the processed cheeses with 0.5% addition. With the additive level increasing, the cheese springiness decreased significantly compared to the control sample. The addition of mulberry caused a significant decrease in cheese springiness compared to the control variant; however, the additive level was found to be of lesser importance as no differences were demonstrated in the springiness values between cheeses with 0.5 and 2.0% mulberry addition.

The results of studies on the introduction of various vegetable additives to processed cheeses indicate some limited possibilities of their use. For example, the addition of 5 to 20% of date fruit seed powder to processed cheese resulted in a deterioration in its texture profile in terms of hardness, adhesiveness, springiness, and cohesiveness, and in lower compactness of the general structure of the cheese, decreasing further with an increasing additive level [38]. Moreover, in the case of our research, the use of an additive in the amount of 1–3% led to significant changes in texture parameters (hardness, adhesiveness, springiness, chewiness). The amount of the tested additives used should be as high as possible due to the increase in the nutritional properties of the product, but, on the other hand, as our research shows, the use of additives in the range of 1–3% can result in both favorable and unfavorable changes in some cheese texture parameters. This makes it difficult to use them in the production of PC because, in small amounts, they may not have health-promoting effects.

Obtaining homogeneous consistency of processed cheese depends on three groups of factors: (i) the composition and properties of the raw materials (type of cheese, degree of maturity, dry matter and fat content, pH of the melting blend, type and concentration of emulsifying salts used, potential addition of hydrocolloids or additives containing substances that affect product consistency); (ii) processing conditions (parameters of mixing and cooling processes); and (iii) storage conditions. In the basic raw material (rennet cheese), the casein fractions are connected by calcium bridges and form a three-dimensional network. The major role of emulsifying salts (mainly sodium polyphosphates) is to exchange calcium from the cheese matrix for sodium ions, owing to which the insoluble Ca-paracaseinate transforms into the more soluble Na-paracaseinate, which can act as an emulsifier and stabilizer. In addition, the use of emulsifying salts affects the emulsification and stabilization of the fat present in the processed cheese matrix, as the protein chains disperse, hydrate, and swell [42]. In the present study, the emulsifying salt content of the processed cheeses was reduced by half owing to their enrichment with 1, 2, and 3% of non-dairy ingredients (Table 1 and Table 2), without compromising their quality characteristics, including consistency. The successful replacement of some of the emulsifying salts allows a reduction in sodium content while developing new products with beneficial health properties.

Although processed cheeses eliciting increased health benefits have recently been addressed in many publications, they are still perceived by consumers as “less healthy” compared to, e.g., ripened cheeses [48]. One of the factors driving consumers’ choices is the product’s price. Reducing the costs of processed cheese production by reducing the content of the most expensive ingredient in the recipe, i.e., rennet cheese, may, on the one hand, be a certain solution for producers [49], but, on the other hand, it may reduce the casein content of the processed cheese, which can negatively affect its texture and flavor. In the present study, the proportion of ripened cheese was reduced by 18% (in the case of a 3% vegetable additive level) in the formulas of processed cheeses with chia seeds and mulberry fruit, compared to the control cheese. This reduction did not result in an undesirable texture profile in the cheeses produced (e.g., in terms of their hardness, cohesiveness, and adhesiveness).

### 3.4. Sensory Quality of Processed Cheeses

Table 5 presents the results of the evaluation of the sensory quality attributes of the analyzed processed cheeses, whereas Figure 1 depicts their appearance. All plant additives diminished the scores given by panelists regarding the appearance of the processed cheeses along with their increasing levels. The lowest score was given to the cheese variants with 1, 2, and 3% addition of green barley and the cheese variant with 3% addition of spirulina. The average score given to the remaining cheese variants for the appearance attribute was over 4.0. The additive level of 0.5% was found not to affect the scores given for the appearance of the cheeses, compared to the control samples, regardless of the additive type.

The taste of the processed cheeses enriched with mulberry did not depend on its content and was equally highly scored as that of the control sample. A significant deterioration in taste, compared to the control sample, was noted in the case of cheeses with 2 and 3% addition of green barley, 1% and higher addition of spirulina, and 3% addition of chia seeds.

The only quality attribute whose score, in most cases, did not depend on the additive level was found to be the aroma, with only one exception, i.e., green barley, whose 1% addition resulted in lower scores given by panelists for this attribute. The increasing levels of the remaining additives in the formulas of processed cheeses had no effect on their aroma evaluation scores.

Among the processed cheese variants enriched with green barley powder, the best-rated variant was the one with 0.5%. It was as attractive to the consumers in terms of taste, aroma, and overall acceptability as the control sample. The cheese samples with 2 and 3% addition of green barley were rated significantly worse in terms of taste (average scores below 3.0) and many evaluators noted that these variants had an unpleasant grassy aftertaste. The achieved biological activity of the green barley additive in processed cheese would be the highest with 3.0% additive, but—according to the present study results—for industrial use, we recommend a maximum additive amount of 1.0%, precisely because of the unpleasant aftertaste. An unbeneficial effect of green barley addition over 3% on pasta taste was reported by Ivanišová et al. [50].

In the work by Tohamy et al. [43], supplemented processed cheese analogs with 2% of *Spirulina platensis* showed the best sensory properties, which were comparable with those of the control sample without the additive, whereas a 6% additive level was found to be unacceptable. These authors also demonstrated that the use of algae powder (similar to the present study) had a less beneficial effect on the organoleptic and rheological properties of cheese analogs than the algae used in the slurry form. In the present study, a significant deterioration in the overall sensory acceptability compared to the control sample was noted even at the lowest level of spirulina addition (0.5%); however, the cheeses were rated below the average score of 4.0 already at 3% algae addition. It seems that the greatest limitation in the use of spirulina as an additive to processed cheese will be its unbeneficial and very poorly accepted impact on product taste at the addition level of more than 0.5%. An unfavorable effect on other sensory attributes was found with the algae inclusion levels of 2 and 3%.

The taste of processed cheeses with 0.5 and 1% addition of chia seeds was rated similarly to that of the control cheeses. The feasibility of supplementing food products with chia seeds, and the lack of adverse effects of this supplementation even at the inclusion level of 10%, was demonstrated by Aja and Haros [25]. In contrast, a reduction in the sensory acceptability of ice cream with 0.8% addition of powdered chia seeds was reported by Ürkek [51]. In turn, Cardoso et al. [9] attempted to enrich processed cheeses with chia seed oil as a source of unsaturated fatty acids and found that its addition resulted in a significant deterioration in cheese taste. The microencapsulation of the chia seed oil masked the undesirable taste to some extent but the overall sensory acceptability of these cheeses was significantly lower compared to the control products.

The processed cheeses with 3.0% mulberry addition were highly rated in terms of aroma, and the panelists emphasized their beneficial “raisin-like” aroma. Hwang and Kim [20] demonstrated that dried mulberry fruit contained benzaldehyde, nonanal, and 3,3-dimethylhexane, i.e., components responsible for grape juice, raisin, and sour aromas. In the present study, the taste of the mulberry-enriched cheeses was rated equally as highly as that of the control samples.

## 4. Conclusions

The processed cheese market is driven by factors such as a growing demand for convenient and ready-to-eat foods and rising consumer incomes, but constrained by health concerns related to the presence of polyphosphates in processed cheese. We have shown that the use of plant-based additives allows us to reduce the addition of emulsifying salts by 50% compared to their typical amounts and the share of rennet cheese in the processed cheese recipe by approximately 18%, which has a beneficial effect on the nutritional value of these products. The use of 3% green barley, chia, or mulberry, as additives to processed cheese, enabled a reduction in its sodium content by 27, 27, and 42%, respectively, compared to the control cheese, by reducing the amounts of emulsifying salts and rennet cheese. Among the tested additives, green barley caused the greatest increase in the hardness of the processed cheese (even at the amount of 0.5%), indicating that it is beneficial and can be used in the production of sliced processed cheese. Plant-based additives also make it possible to expand the development of processed cheeses with new sensory properties (color or taste). Among the analyzed additives, the most beneficial effect on the sensory properties of processed cheeses, as well as other evaluated parameters, was with the addition of white mulberry fruit. The use of green barley and spirulina at levels higher than 1% caused an unpleasant aftertaste in the processed cheeses.

As mankind develops, the need for physical effort decreases, which means that our daily caloric demands should also decrease, but this has not happened. With age, the energy requirement of a human decreases by approximately 40–50%, but the daily ritual of eating meals does not change significantly. Taking into account these two phenomena, the populations of highly developed countries face problems related to lifestyle-related diseases, including type II diabetes, cardiovascular diseases, and cancer. Hence, the changes that need to be made should mainly concern our ways of thinking. The food industry may respond to these phenomena by offering products with reduced energy value, i.e., with reduced fat, sugar, and protein content. This is feasible through the use of appropriate additives. In addition, the use of some plant additives significantly enriches the composition and nutritional value of processed cheeses and at the same time allows a reduction in the addition of polyphosphate emulsifying salts, which have an adverse effect on their nutritional value.

## Figures and Tables

**Figure 1 foods-12-02862-f001:**
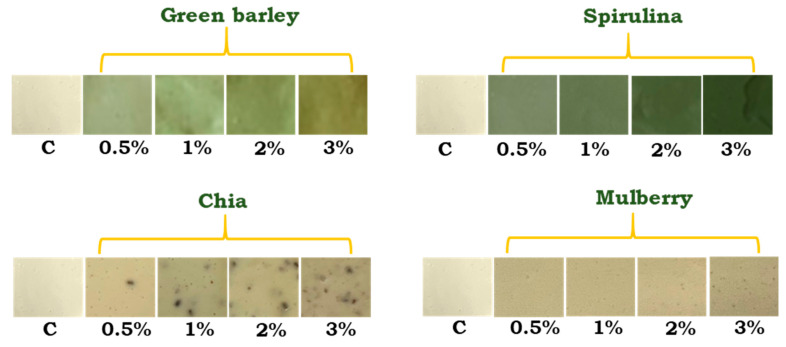
Photos of samples of processed cheeses with 0.5, 1, 2, and 3% addition of green barley, spirulina, chia seeds, and mulberry fruit, and a control sample without the plant additive (C).

**Figure 2 foods-12-02862-f002:**
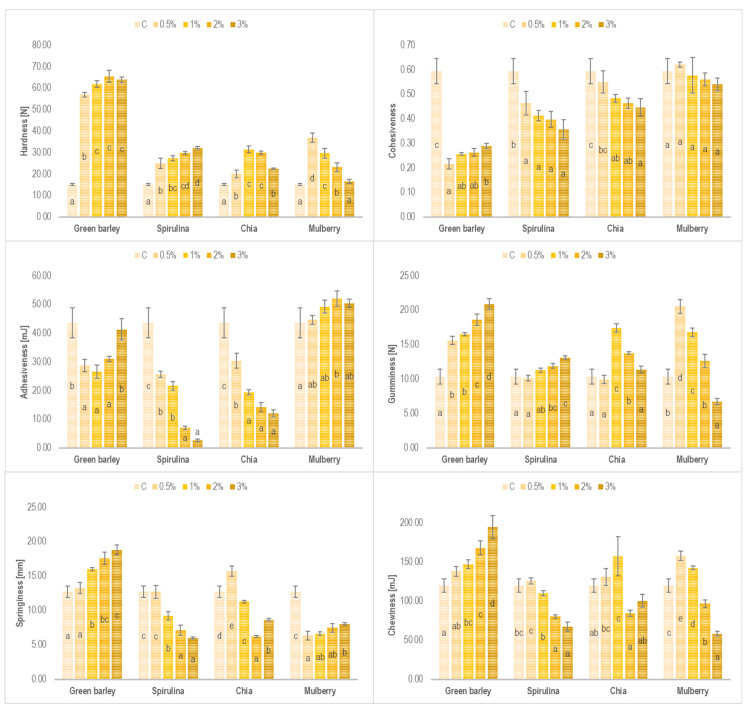
Texture parameters (hardness, cohesiveness, adhesiveness, gumminess, springiness, and chewiness) of processed cheeses with 0.5, 1, 2, or 3% addition of green barley, spirulina, chia seeds, or mulberry, and a control sample without the additive; C—control sample; mean values ± standard deviation; the means within the same additive followed by different lowercase letters differ from each other (*p* < 0.05).

**Table 1 foods-12-02862-t001:** Formulas of the processed cheeses produced with the addition of green barley and spirulina.

Ingredients	Processed Cheese Variant				
	Control	Amount of Additive [g]			
		0.5%	1.0%	2.0%	3.0%
Rennet ripening cheese	560	560	560	555	550
Water	320	320	320	315	310
Butter	100	100	100	100	100
Emulsifying salts	20	20	10	10	10
Additive	0	5	10	20	30

**Table 2 foods-12-02862-t002:** Formulas of the processed cheeses produced with the addition of chia seeds and white mulberry fruit.

Ingredients	Processed Cheese Variant				
	Control	Amount of Additive [g]			
		0.5%	1.0%	2.0%	3.0%
Rennet ripening cheese	560	555	540	510	460
Water	320	320	340	360	400
Butter	100	100	100	100	100
Emulsifying salts	20	20	10	10	10
Plant additive	0	5	10	20	30

**Table 3 foods-12-02862-t003:** Results of the chemical and physicochemical analysis of processed cheeses with 0.5, 1, 2, or 3% addition of green barley (GB), spirulina (S), chia seeds (Ch), and mulberry (M), and of the control sample without the plant additive; mean values ± standard deviation.

Processed Cheese Variant		*DM* (%)	Fat (%)	*FDM* (%)	Protein (%)	Sodium (%)	pH	a_w_
Control		42.88b ± 1.20	22.53d ± 0.13	52.57c ± 1.32	15.05a ± 0.10	1.06b ± 0.16	5.96b ± 0.01	0.916a ± 0.004
GB	0.5%	44.15ab ± 0.02	21.21c ± 0.03	48.04b ± 0.07	16.59b ± 0.05	1.04b ± 0.03	5.99b ± 0.02	0.934b ± 0.002
	1%	45.26a ± 0.03	21.15c ± 0.03	46.74ab ± 0.08	16.97c ± 0.04	0.81a ± 0.04	5.96b ± 0.01	0.936b ± 0.001
	2%	44.85a ± 0.02	20.73b ± 0.04	46.22a ± 0.08	17.24d ± 0.03	0.79a ± 0.04	5.86a ± 0.05	0.937b ± 0.003
	3%	43.13b ± 0.05	19.78a ± 0.02	45.87a ± 0.09	16.60b ± 0.01	0.77a ± 0.02	5.81a ± 0.01	0.940b ± 0.002
Control		42.88c ± 1.20	22.53c ± 0.13	52.57b ± 1.32	15.05bc ± 0.10	1.06b ± 0.16	5.96d ± 0.01	0.916a ± 0.004
S	0.5%	45.83a ± 0.23	22.47c ± 0.10	49.02a ± 0.46	15.55d ± 0.09	1.07b ± 0.05	6.00d ± 0.03	0.944b ± 0.004
	1%	45.40ab ± 0.42	21.98b ± 0.10	48.41a ± 0.23	15.29cd ± 0.19	0.85a ± 0.06	5.77a ± 0.03	0,952c ± 0.002
	2%	44.06bc ± 0.07	21.26a ± 0.13	48.25a ± 0.23	14.73b ± 0.10	0.83a ± 0.06	5.83b ± 0.02	0.954cd ± 0.002
	3%	40.54d ± 0.41	21.01a ± 0.11	51.81b ± 0.32	13.93a ± 0.14	0.86a ± 0.03	5.90c ± 0.01	0.960d ± 0.002
Control		42.88b ± 1.20	22.53a ± 0.13	52.57ab ± 1.32	15.05c ± 0.10	1.06d ± 0.16	5.96d ± 0.01	0.916a ± 0.004
Ch	0.5%	45.13a ± 0.11	23.68b ± 0.54	52.48ab ± 1.26	15.44cd ± 0.20	1.05d± 0.02	6.01e ± 0.01	0.933bc ± 0.010
	1%	46.00a ± 0.10	23.76b ± 0.37	51.66a ± 0.91	15.82d ± 0.35	0.90c ± 0.04	5.77a ± 0.02	0.927ab ± 0.002
	2%	40.64c ± 0.33	22.44a ± 0.41	55.22b ± 1.43	14.41b ± 0.18	0.84b ± 0.04	5.83b ± 0.02	0.944cd ± 0.003
	3%	37.86d ± 0.16	22.49a ± 0.09	59.40c ± 0.15	13.44a ± 0.13	0.77a ± 0.03	5.90c ± 0.02	0.957d ± 0.003
Control		42.88b ± 1.20	22.53c ± 0.13	52.57b ± 1.32	15.05d ± 0.10	1.06d ± 0.16	5.96d ± 0.01	0.916a ± 0.004
M	0.5%	45.51a ± 0.58	23.42d ± 0.09	51.46ab ± 0.79	15.50e ± 0.04	1.02d ± 0.04	5.81c ± 0.01	0.943b ± 0.002
	1%	44.55a ± 0.06	22.74c ± 0.04	51.05ab ± 0.06	14.71c ± 0.04	0.97c ± 0.02	5.57a ± 0.01	0.944b ± 0.002
	2%	41.95b ± 0.34	21.25b ± 0.06	50.67a ± 0.36	13.61b ± 0.04	0.79b ± 0.05	5.58ab ± 0.01	0.946b ± 0.004
	3%	40.27c ± 0.07	20.12a ± 0.05	49.98a ± 0.05	12.62a ± 0.04	0.61a ± 0.03	5.60b ± 0.01	0.946b ± 0.002

*DM*—dry matter content, *FDM*—fat in dry matter content; the means within the columns (within the same type of additive) followed by different lowercase letters differ from each other (*p* < 0.05).

**Table 4 foods-12-02862-t004:** Color parameters of processed cheeses with 0.5, 1, 2, or 3% addition of green barley (GB), spirulina (S), chia seeds (Ch), or mulberry (M), and a control sample without the plant additive, mean values ± standard deviation.

Processed Cheese Variant		*L**	*a**	*b**	∆*E*
Control		92.31 ± 0.02	1.06 ± 0.01	15.13 ± 0.02	-
GB	0.5%	79.57d ± 0.03	−5.69d ± 0.00	23.80a ± 0.05	16.80a ± 0.04
	1%	74.31c ± 0.02	−5.97b ± 0.04	26.32b ± 0.12	22.33b ± 0.07
	2%	67.85b ± 0.10	−6.13a ± 0.02	29.42c ± 0.15	29.22c ± 0.01
	3%	62.98a ± 0.05	−5.78c ± 0.01	30.10d ± 0.02	33.63d ± 0.04
S	0.5%	71.53d ± 0.01	−6.50d ± 0.02	16.75a ± 0.04	22.17a ± 0.00
	1%	65.82c ± 0.04	−7.22b ± 0.02	17.49c ± 0.02	27.85b ± 0.03
	2%	57.96b ± 0.02	−7.32a ± 0.02	18.10d ± 0.04	35.48c ± 0.02
	3%	52.92a ± 0.03	−7.02c ± 0.01	17.02b ± 0.07	40.26d ± 0.03
Ch	0.5%	89.06d ± 0.14	1.14a ± 0.02	13.43d ± 0.06	3.67a ± 0.15
	1%	86.04c ± 0.19	1.21b ± 0.02	12.87c ± 0.02	6.66b ± 0.17
	2%	83.68b ± 0.09	1.26b ± 0.03	12.38b ± 0.08	9.06c ± 0.06
	3%	80.72a ± 0.15	1.43c ± 0.04	12.00a ± 0.05	12.01d ± 0.13
M	0.5%	90.84d ± 0.01	1.11a ± 0.02	15.35c ± 0.02	1.49a ± 0.01
	1%	90.66c ± 0.06	1.25b ± 0.03	14.38a ± 0.04	1.82b ± 0.05
	2%	89.73b ± 0.02	1.39c ± 0.01	14.54b ± 0.04	2.67c ± 0.01
	3%	87.85a ± 0.05	1.77d ± 0.01	15.86d ± 0.02	4.58d ± 0.04

∆*E*—total color difference between processed cheeses with additive and control sample without additive; the means within the columns (within the same type of additive) followed by different lowercase letters differ from each other (*p* < 0.05).

**Table 5 foods-12-02862-t005:** Results of organoleptic assessment of processed cheeses with 0.5, 1, 2, or 3% addition of green barley (GB), spirulina (S), chia seeds (Ch), or mulberry (M), and a control sample without the additive; mean values ± standard deviation.

Processed Cheese Variant		Appearance	Taste	Aroma	Consistency	Overall Acceptability
GB	Control	5.00c ± 0.00	4.80c ± 0.35	4.90b ± 0.21	4.90c ± 0.21	4.90c ± 0.13
	0.5%	4.95c ± 0.16	4.20c ± 0.54	4.45b ± 0.64	4.50bc ± 0.58	4.53c ± 0.44
	1%	3.85b ± 0.82	3.80bc ± 0.79	3.20a ± 0.79	3.85b ± 0.88	3.68b ± 0.36
	2%	3.30ab ± 0.75	2.85ab ± 1.00	3.30a ± 0.59	3.05a ± 0.55	3.13a ± 0.18
	3%	3.00a ± 0.88	2.70a ± 1.09	3.15a ± 0.58	2.70a ± 0.67	2.89a ± 0.25
S	Control	5.00c ± 0.00	4.80b ± 0.35	4.90a ± 0.21	4.90b ± 0.21	4.90c ± 0.13
	0.5%	4.80bc ± 0.35	4.15ab ± 0.47	4.65a ± 0.34	4.30ab ± 0.63	4.48b ± 0.36
	1%	4.75bc ± 0.35	3.85a ± 0.63	4.65a ± 0.41	4.65b ± 0.41	4.48b ± 0.26
	2%	4.19ab ± 0.48	3.50a ± 0.75	4.55a ± 0.50	4.15ab ± 0.78	4.10ab ± 0.43
	3%	3.85a ± 0.94	3.45a ± 0.64	4.35a ± 0.67	3.75a ± 0.72	3.85a ± 0.37
Ch	Control	5.00b ± 0.00	4.80b ± 0.35	4.90a ± 0.21	4.90c ± 0.21	4.90c ± 0.13
	0.5%	4.85b ± 0.34	4.50ab ± 0.53	4.80a ± 0.35	4.70bc ± 0.35	4.71bc ± 0.20
	1%	4.60ab ± 0.46	4.50ab ± 0.47	4.75a ± 0.42	4.10ab ± 0.70	4.49abc ± 0.34
	2%	4.35ab ± 0.67	3.95a ± 0.69	4.40a ± 0.70	3.75a ± 0.68	4.11a ± 0.35
	3%	4.10a ± 0.77	4.00a ± 0.71	4.65a ± 0.41	4.55bc ± 0.50	4.33ab ± 0.56
M	Control	5.00c ± 0.00	4.80a ± 0.35	4.90a ± 0.21	4.90b ± 0.21	4.90b ± 0.13
	0.5%	4.80bc ± 0.26	4.45a ± 0.50	4.95a ± 0.16	4.95b ± 0.16	4.79b ± 0.08
	1%	4.45ab ± 0.28	4.60a ± 0.21	4.90a ± 0.32	4.15a ± 0.34	4.53a ± 0.16
	2%	4.40a ± 0.39	4.60a ± 0.32	4.85a ± 0.34	4.35a ± 0.41	4.55a ± 0.15
	3%	4.30a ± 0.42	4.40a ± 0.39	4.90a ± 0.32	4.30a ± 0.26	4.48a ± 0.17

The means within the columns (within the same type of additive) followed by different lowercase letters differ from each other (*p* < 0.05).

## Data Availability

Data is contained within the article.

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
