# Peer review of "The Impact of White Mulberry, Green Barley, Chia Seeds, and Spirulina on Physicochemical Characteristics, Texture, and Sensory Quality of Processed Cheeses"

_foods, 2023, doi:10.3390/foods12152862_

Round 1

Reviewer 1 Report

Comments and Suggestions for Authors

RESPONSE TO FOODS-2524977

THE IMPACT OF WHITE MULBERRY, GREEN BARLEY, CHIA SEEDS, AND SPIRULINA ON PHYSICOCHEMICAL CHARACTERISTICS, TEXTURE, AND SENSORY QUALITY OF PROCESSED CHEESE

The findings of this study are highly relevant in the context of food processing and beneficial to understand the effects of plant-additives on processed cheese with aim to produce processed cheese with high health benefits. The manuscript's content is commendable, and I was able to comprehend and interpret the data effectively. However, there are several areas that require amendments. Please respond to the following comments;

Abstract

1. “In all variants of PC…..” and “Each of the tested additives…” – was the changes significant?

2. Add this work contribution for the food industry of processed cheese.

Introduction

1. Revise these sentences that are seem redundant with too many “ands” – “They are produces by grinding….”  and “Ingredients are mixed and heated for a…”

2. Revise – “Improved novel products could also…”

3. Omit this sentence – “Processed cheese are considered good…”

Materials and Methods

1. What was the basis of your formulations? Past studies? Your own study? Industrial recipe?

2. Measurement of water activity – reference?

3. Acidity measurement - Reference?

4. Measurement of color – reference?

5. Texture analysis – reference?

Results and Discussion

1. DM?

2. “In all variants of processed cheese, a decrease in DM…” – was the decreased significant?

3. “A similar correlation was found in the case of fat content…” – why?

4. “this indicates that different amount of protein were introduced into the processed cheese…” – revise this sentence. It was difficult to understand.

5. “The sodium content of PC decreased statistically significantly..” – revise

6. ΔE can be compared to the control sample via Dunnet’s comparison test to seek the significance of plant-additives addition to the PC from the control sample.

7. The paragraph that contains “The results of studies on the introduction of various vegetable additives to PC…” seems relevant to the introduction section. Perhaps you could revise to include the results of your work that would produce disadvantageous/limiting effects to plant-additives added PC.

8. “un unbeneficial effect of green barley…” -- ??

9. This section would benefit from a more substantial analysis and should be enhanced. It is recommended to include comparisons to other plant-additives addition to PC and analyses of the findings with those.

Conclusion

1. To revise. The conclusion section should address/answer the objectives of this work. Highlighting the results of those that are significance and important to PC producers.

References:

1. All references are relevant to the study.

I recommend that the minor modifications to the article provided that thorough revision and justification are done properly.

Comments on the Quality of English Language

Minor corrections are needed. 

Author Response

Dear Reviewer,

Thank you very much for reviewing our manuscript: “The impact of white mulberry, green barley, chia seeds, and spirulina on physicochemical characteristics, texture, and sensory quality of processed cheeses”. We have adopted all Your suggestions.

Your suggestions have seriously contributed to the improvement of our manuscript. All changes compared to the original version have been highlighted in green. Hope the revised manuscript will be evaluated as improved, in any case, we are willing to consider any further request.

 SPECIFIC COMMENTS

Abstract

  1. “In all variants of PC…..” and “Each of the tested additives…” – was the changes significant?

Changed in modified text

  1. Add this work contribution for the food industry of processed cheese.

The results of this research may be helpful in the development of new recipes for processed cheeses obtained in industrial conditions.

Changed in modified text

Introduction

  1. Revise these sentences that are seem redundant with too many “ands” – “They are produces by grinding….” and “Ingredients are mixed and heated for a…”

Changed in modified text

  1. Revise – “Improved novel products could also…”

Changed in modified text

  1. Omit this sentence – “Processed cheese are considered good…”

Changed in modified text

Materials and Methods

  1. What was the basis of your formulations? Past studies? Your own study? Industrial recipe?

The processed cheese formula was developed based on our previous research (unpublished data).

  1. Measurement of water activity – reference?

Changed in modified text

  1. Acidity measurement - Reference?

Changed in modified text

  1. Measurement of color – reference?

 Changed in modified text

  1. Texture analysis – reference?

Changed in modified text

Results and Discussion

1.DM?

We added the explanation of abbreviation in subchapter 2.3.

2.“In all variants of processed cheese, a decrease in DM…” – was the decreased significant?

Changed in modified text

  1. “A similar correlation was found in the case of fat content…” – why?

Changed in modified text

  1. “this indicates that different amount of protein were introduced into the processed cheese…” – revise this sentence. It was difficult to understand.

Changed in modified text

  1. “The sodium content of PC decreased statistically significantly..” – revise

Changed in modified text

  1. ΔE can be compared to the control sample via Dunnet’s comparison test to seek the significance of plant-additives addition to the PC from the control sample.

Changed in modified text – modification and statistical analysis in Table 4

  1. The paragraph that contains “The results of studies on the introduction of various vegetable additives to PC…” seems relevant to the introduction section. Perhaps you could revise to include the results of your work that would produce disadvantageous/limiting effects to plant-additives added PC.

Changed in modified text

  1. “un unbeneficial effect of green barley…” -- ??

Changed in modified text

  1. This section would benefit from a more substantial analysis and should be enhanced. It is recommended to include comparisons to other plant-additives addition to PC and analyses of the findings with those.

Thank you very much for your comment, we agree that additional discussion would enrich this chapter, but there is not much literature data in this research field. We reviewed the available literature again and made some changes where possible.

Conclusion

To revise. The conclusion section should address/answer the objectives of this work. Highlighting the results of those that are significance and important to PC producers.

The section „Conclusions” was revised.

Reviewer 2 Report

Comments and Suggestions for Authors

I am very grateful you for the invitation to review manuscript foods-2524977 by Garbowska and coauthors "The impact of white mulberry, green barley, chia seeds, and spirulina on physicochemical characteristics, texture, and sensory quality of processed cheeses”. The aim of the present study was to determine the possibility of using functional ingredients, such as white mulberry, chia seeds, green barley, and spirulina, to produce processed cheeses, and to examine the effect of these additives on their physicochemical composition, water activity, rheology, and sensory properties. The work is interesting but needs adjustments to increase the quality of the material.

Comments:

- Abstract, Processed cheeses (PC): Define this product type in the first presentation.

- Abstract: “can also affect the quality characteristics of PC (e.g., texture)”: Generic information. Specify whether it affects positively, negatively, etc.

- Abstract: Please indicate a better step-by-step about the work.

- Abstract, “Each of the tested additives affected the hardness, and cohesiveness of PC.”: Positively or negatively. The information is not clear.

- Abstract, “The emulsifying salts content of PC was reduced by half owing to their enrichment with 1, 2, or 3% of ingredients”: Which added component resulted in this modification?

-  Abstract: The authors do not present a conclusion about the study.

- Keywords: Change the repeated keywords by different words from the title

- Introduction: Technological aspects and interactions during cheese processing must be presented since ingredients cause changes in these interactions.

- Introduction: Include recent data regarding the processed cheeses market.

- Introduction: What is the basic nutritional composition of this type of product? This should be clear, as nutrient enrichment is highlighted.

- Introduction, “plants as sources of various biologically active substances”: What substances have been used? Specify better.

- Introduction, “The processing of plants, such as white mulberry, green barley and chia, is an interesting and useful task”: Chia is being widely used.

- Introduction: Specify the world production of each of the highlighted raw materials, presenting the potential and feasibility of use.

- Introduction: Presentation of nutritional components, including concentration, should be improved. The authors did not present the content of any component of Mulberry, for example.

- Standardize throughout the text the use of the terms “fluxing agent used”, “emulsifying salts used” and others.

- Standardize font style and size across all text.

- 2.5. Acidity measurements: Acidity and pH are different parameters. Correct the title of the item, as acidity was not measured.

- Results, “In all variants of processed cheeses, a decrease in DM content was

observed as the additive level increased”: This sentence is not clear. Please specify better.

- Results, “The active acidity (pH) of the”: Although they are correlated, they are different measurements. Please change throughout the work, as there are specific methodologies for quantifying acidity in cheeses.

- Results: The authors must correlate the presence of components with the technological properties of the cheese. It only informs if there was a reduction or increase in levels is not enough.

- 3.2. Color parameters of processed cheeses: This item is also generic and must be related to pigments found in additives used in processed cheese.

- 3.3. Texture profile of processed cheese samples: This item must be improved with the specification of alterations caused by the addition of additives and, mainly, by the components that alter the structure of the processed cheese.

- Conclusion: This item is not a conclusion. Authors must rewrite the conclusion presenting the main information obtained based on the objective of the work.

Author Response

Dear Reviewer,

Thank you very much for reviewing our manuscript: “The impact of white mulberry, green barley, chia seeds, and spirulina on physicochemical characteristics, texture, and sensory quality of processed cheeses”. We have adopted all Your suggestions.

Your suggestions have seriously contributed to the improvement of our manuscript. All changes compared to the original version have been highlighted in blue (the second reviewer - in green). Hope the revised manuscript will be evaluated as improved, in any case, we are willing to consider any further request.

SPECIFIC COMMENTS

- Abstract, Processed cheeses (PC): Define this product type in the first presentation.

Changed in modified text

- Abstract: “can also affect the quality characteristics of PC (e.g., texture)”: Generic information. Specify whether it affects positively, negatively, etc.

Changed in modified text

- Abstract: Please indicate a better step-by-step about the work.

Changed in modified text

- Abstract, “Each of the tested additives affected the hardness, and cohesiveness of PC.”: Positively or negatively. The information is not clear.

Changed in modified text

- Abstract, “The emulsifying salts content of PC was reduced by half owing to their enrichment with 1, 2, or 3% of ingredients”: Which added component resulted in this modification?

Changed in modified text

-  Abstract: The authors do not present a conclusion about the study.

Changed in modified text

- Keywords: Change the repeated keywords by different words from the title

Changed in modified text

- Introduction: Technological aspects and interactions during cheese processing must be presented since ingredients cause changes in these interactions.

This information is given in chapter 3.3. starting from: “Obtaining a homogeneous consistency of processed cheese depends on three groups of factors: (i) the composition and properties ….”

- Introduction: Include recent data regarding the processed cheeses market.

Changed in modified text

- Introduction: What is the basic nutritional composition of this type of product? This should be clear, as nutrient enrichment is highlighted.

Changed in modified text

- Introduction, “plants as sources of various biologically active substances”: What substances have been used? Specify better.

Changed in modified text

- Introduction, “The processing of plants, such as white mulberry, green barley and chia, is an interesting and useful task”: Chia is being widely used.

Changed in modified text

- Introduction: Specify the world production of each of the highlighted raw materials, presenting the potential and feasibility of use.

We did not find data of this type in the FAOSTAT database. In addition, data on the world production volume of the individual additives we investigate do not seem to us necessary for this scope of our research.

- Introduction: Presentation of nutritional components, including concentration, should be improved. The authors did not present the content of any component of Mulberry, for example.

Changed in modified text

- Standardize throughout the text the use of the terms “fluxing agent used”, “emulsifying salts used” and others.

Changed in modified text

- Standardize font style and size across all text.

Changed in modified text

- 2.5. Acidity measurements: Acidity and pH are different parameters. Correct the title of the item, as acidity was not measured.

Changed in modified text

- Results, “In all variants of processed cheeses, a decrease in DM content was observed as the additive level increased”: This sentence is not clear. Please specify better.

Changed in modified text

- Results, “The active acidity (pH) of the”: Although they are correlated, they are different measurements. Please change throughout the work, as there are specific methodologies for quantifying acidity in cheeses.

Changed in modified text

- Results: The authors must correlate the presence of components with the technological properties of the cheese. It only informs if there was a reduction or increase in levels is not enough.

Changed in modified text

- 3.2. Color parameters of processed cheeses: This item is also generic and must be related to pigments found in additives used in processed cheese.

Changed in modified text

- 3.3. Texture profile of processed cheese samples: This item must be improved with the specification of alterations caused by the addition of additives and, mainly, by the components that alter the structure of the processed cheese.

Changed in modified text

- Conclusion: This item is not a conclusion. Authors must rewrite the conclusion presenting the main information obtained based on the objective of the work.

The section “Conclusions” was revised.

Round 2

Reviewer 2 Report

Comments and Suggestions for Authors

The quality of the manuscript is improved.